# Assessment of Ionomic, Phenolic and Flavonoid Compounds for a Sustainable Management of *Xylella fastidiosa* in Morocco

**Kaoutar El Handi [1,*], Majida Hafidi [2], Khaoula Habbadi [1], Maroun El Moujabber [3], Mohamed Ouzine [1], Abdellatif Benbouazza [1], Miloud Sabri [1] and El Hassan Achbani [1]**

1   Laboratory of Phyto-Bacteriology and Biocontrol, Plant Protection Unit-National Institute of Agronomic Research INRA, Meknès 50000, Morocco; khaoula405@gmail.com (K.H.); mohamed.ouzine1@usmba.ac.ma (M.O.); dacus5@hotmail.com (A.B.); miloud.sabri@uit.ac.ma (M.S.); achbaniofficiel@gmail.com (E.H.A.)
2   Laboratory of Biology, Moulay Ismail University, Meknès 50000, Morocco; hafidimaj@yahoo.fr
3   CIHEAM Bari, Istituto Agronomico Mediterraneo, Via Ceglie 9, Valenzano, 70010 Bari, Italy; elmoujabber@iamb.it
*   Correspondence: kaoutar.elhandi@edu.umi.ac.ma

**Abstract:** Morocco belongs to the countries ranked at a high-risk level for entry, establishment, and spread of *Xylella fastidiosa*, which has recently re-emerged as a plant pathogen of global importance causing olive quick decline syndrome (OQDS). Symptomatic infection by *X. fastidiosa* leads to devastating diseases and important economic losses. To prevent such losses and damages, countries without current outbreaks like Morocco need to first understand their host plant responses to *X. fastidiosa*. The assessment of the macro and micro-elements content (ionome) in leaves can give basic and useful information along with being a powerful tool for the sustainable management of diseases caused by this devastating pathogen. Herein, we compare the leaf ionome of four important autochthonous Moroccan olive cultivars ('Picholine Marocaine', 'Haouzia', 'Menara', and 'Meslalla'), and eight Mediterranean varieties introduced in Morocco ('Arbequina', 'Arbosana', 'Leccino', 'Ogliarola salentina', 'Cellina di Nardo', 'Frantoio', 'Leucocarpa', and 'Picholine de Languedoc'), to develop hypotheses related to the resistance or susceptibility of the Moroccan olive trees to *X. fastidiosa* infection. Leaf ionomes, mainly Ca, Cu, Fe, Mg, Mn, Na, Zn, and P, were determined using inductively coupled plasma optical emission spectroscopy (ICP-OES). These varieties were also screened for their total phenolics and flavonoids content. Data were then involved in a comparative scheme to determine the plasticity of the pathogen. Our results showed that the varieties 'Leccino', 'Arbosana', 'Arbequina' consistently contained higher Mn, Cu, and Zn and lower Ca and Na levels compared with the higher pathogen-sensitive 'Ogliarola salentina' and 'Cellina di Nardò'. Our findings suggest that 'Arbozana', 'Arbiquina', 'Menara', and 'Haouzia' may tolerate the infection by *X. fastidiosa* to varying degrees, provides additional support for 'Leccino' having resistance to *X. fastidiosa*, and suggests that both 'Ogliarola salentina' and 'Cellina di Nardö' are likely sensitive to *X. fastidiosa* infection.

**Keywords:** olive quick decline syndrome; *Xylella fastidiosa*; calcium; manganese; Leccino; Leccinola salentina; olive; Moroccan olive varieties; Mediterranean olive varieties

## 1. Introduction

In Morocco, olive (*Olea europaea* subsp. *europaea* L.) groves have a crucial socio-economic role, representing the main source of livelihood for many local farmers. Moroccan olive groves represent the Southwesternmost part of the Mediterranean olive-growing landscape. In this country, olive cultivation and oil production are a deep-rooted tradition, both as an income for more than 450,000 farmers and a high environmental value crop, due to its role in soil protection, particularly, in mountain farms [1]. Furthermore, over the last few years, land use for olive cultivation in Morocco has increased from 946,818 ha

in 2014 to 1,073,493 ha in the 2019 growing season [2] making this crop one of the most profitable and strategic horticultural crops in the country. The 'Picholine Marocaine' is the predominant variety; more than 96% of olive groves are planted with this variety [3]. Two varieties, 'Menara' and 'Haouzia', registered for cultivation in Morocco, were developed through clonal selection [4].

However, despite its importance, the olive crop presents some constraints, especially related to biotic stresses caused by plant pathogens [5]. *X. fastidiosa* is an important plant pathogen that attacks several economically important plants including the olive tree [6,7]. This pathogen has a very wide host range, including plants belonging to 595 plant species, 275 genera and 85 families [8]; however, it is well known as the causal agent of grapevine Pierce's disease (PD) and of Citrus Variegated Chlorosis (CVC) in South and North America. In Europe, *X. fastidiosa* is regulated as a quarantine pathogen; only individual outbreaks were reported until 2013 when the bacterium was detected for the first time in the Southern part of the Apulia region (Southern Italy) [8], one of the major olive-growing areas in Italy. The disease, named Olive Quick Decline Syndrome (OQDS), has a highly destructive impact on the infected trees and is characterized by leaf scorching, desiccation of leaves, twigs and branches and leads the whole tree to death within few years [9]. The olive quick decline syndrome (OQDS) caused by *X. fastidiosa* is one of the most damaging diseases threatening olive trees worldwide [9].

Studies on this disease are progressing, but quantifiable data and estimates on its spread are scarce [8]. To date, Morocco is considered a country free from this devastating bacterium [10]. However, the likelihood of *X. fastidiosa* occurrence and spread in Morocco is high due to the ineffective control measures adopted, and to the insect vectors that transmit the bacterium across the world [10]. Indeed, Morocco is 13 km from Spain and is the gateway to all African countries. Also, Morocco has significant trade with several European countries, notably France, Spain, and Italy, which may contribute to the introduction of *X. fastidiosa* indirectly via infected plant material [10].

The OQDS is characterized by leaf scorching and scattered desiccation of twigs and small branches which, in the early stages of the infection, are observed on the upper part of the canopy [11]. Leaf tips and margins turn from dark yellow to brown and eventually dry out. Over time, symptoms increase severely and extend to the rest of the canopy, which acquires a blighted appearance [12]. According to D'Attoma et al. [13], variation in leaf mineral nutrients is also associated with the olive infection by *X. fastidiosa*.

The analysis of the whole profile of trace elements and mineral nutrients can contribute to the evaluation the physiological status of the plant in inter-connection with the pathogen infection [14]. Studies on mineral elements accumulation in specific plant tissues, especially in the leaves, have been used to assess the physiological status of the plant [13]. Regarding disease development in plant hosts, the influence of specific mineral elements is well documented. However, the ionome has only been used in a few instances as a composite phenotypic character to assess the relationship between plants and *X. fastidiosa* infection. Indeed, some studies understandably indicated a correlation between the content of some ions in the leaf and the virulence of *X. fastidiosa* [13,15]. It was highly recommended that the host ionome and its variation could be considered as a potential tool for the control of diseases caused by this xylem-limited phytopathogenic bacterium [13]. Mineral transport and balance are crucial for the growth and development of plants and also microorganisms and can be a major factor in disease control and progression [16]. In host–pathogen interactions, the competition for these elements is a phenomenon known as "nutritional immunity" [17]. The ionome analysis, defined as the total profile of the mineral nutrients and trace elements found in an organism, represents a pathogen's approach to looking into the physiological status of the plant [18]. In previous studies, the ionome analyses of field-grown blueberry, pecan, grapevines, and greenhouse-grown tobacco during *X. fastidiosa* infection, highlighted significant changes between uninfected and infected plants, as well as between symptomatic and asymptomatic leaves, revealing a complex interaction between different elements in the host [14,19,20]. *X. fastidiosa* accumulates high levels of metals (Mn,

Zn, and Cu) in its biofilm cells (important for the virulence of this bacterium), as compared with planktonic cells [21]. Additionally, several elements (Ca, Mg, and Fe) are known to promote the expression of virulence traits [15,22,23] whereas others (Cu and Zn) have a deleterious effect on growth and biofilm production [24]. Particular attention has been given also to the identification of secondary metabolites that are essential for plant disease resistance [25] as possible strategies for disease management. Bacterial growth, assemblage and biofilm formation could be affected by xylem sap components [15]. Phenolic acids and flavonoids have been shown to inhibit the in vitro growth of *X. fastidiosa* [25].

The aim of the present work was to find a sustainable management tool for olive trees threatened by *X. fastidiosa*. The tool consists in a thorough determination and comparison of the ionomic, phenolics and flavonoids profile of 'Picholine Marocaine', the most widespread and typical Moroccan olive variety, with those of Moroccan clonal selected varieties ('Haouzia', 'Menara', and 'Meslalla') and eight Mediterranean varieties recently introduced in Morocco ('Arbequina', 'Arbosana', 'Leccino', 'Ogliarola', 'Cellina di Nardo', 'Frantoio', 'Leucocarpa' and 'Picholine de Languedoc') to develop hypotheses related to the resistance or susceptibility of the Moroccan olive trees to *X. fastidiosa* infection. All the varieties were grown in the same experimental olive grove and under identical pedoclimatic conditions. To the best of our knowledge, we will be the first in Morocco to carry out this type of proactive research on *X. fastidiosa*. However, this research is easily replicable in all Mediterranean countries where olive trees are under an existential pathogenic threat.

## 2. Materials and Methods

### 2.1. Samples Collection

In order to assess the leaf ionomic profiles of four autochthonous Moroccan cultivars ('Picholine Marocaine', 'Haouzia', 'Menara', and 'Meslalla'), and eight Mediterranean varieties introduced in Morocco ('Arbequina', 'Arbosana', 'Leccino', 'Ogliarola', 'Cellina di Nardo', 'Frantoio', 'Leucocarpa' and 'Picholine de Languedoc'), leaf samples were collected from uninfected 16-year-old olive trees in late December 2020 from olive groves located at the experimental station of the National Agricultural Research Institute (INRA) in Ain Taoujdate, Fez-Meknes region (Morocco). For each tree, five branches were selected, and mature leaves were detached from the median part of hardwood cuttings and collected for analysis. All the olive trees were grown in the same experimental olive grove and under identical pedoclimatic conditions.

### 2.2. Polymerase Chain Reaction

Leaf samples were tested by PCR for the detection of *X. fastidiosa*. Total DNA was extracted from leaf petioles and midveins using a CTAB- based extraction buffer [24,26]. For PCR, the RST31/RST33 set of primers targeting the 16 S rDNA gene was used [26]. The used primers (RST31/RST33) are widely accepted for the detection of the bacterium in quarantine programs, as well as primers targeting the 16 S rDNA genomic region, which are more acceptable for the precise detection of a wider number of genetically diverse strains of *X. fastidiosa*. Reactions were conducted in a final volume of 20 µL, using 0.5 µL each of forward and reverse primer, 3 µL of total DNA template and 3 µL of 5× GoTaq polymerase (Promega). PCR was completed using a (5 PRIMEG/C Serial No 51147-2) thermocyler set to the following: 94 °C for 3 min, 35 cycles of 94 °C for 30 s, 50–55 °C for 30–45 s and 72 °C for 30 s, and a final extension of 5 min at 72 °C [27]. The resulting PCR products were visualized by electrophoresis in 1% Tris-Acetate-EDTA agarose gel stained with ethidium bromide.

### 2.3. Extract Preparation for Assessment of Total Phenolic and Flavonoid Content

Leaves were cut and frozen at −20 °C for later lyophilization. Then, they were ground into powder at room temperature using an IKA A11 Basic Grinder (St. Louis, MO, USA). Extraction was based on the method previously described by Sanders et al. [28] and moderately modified by Xie and Bolling [29]. First, 1 g aliquots of powder were

transferred into polypropylene tubes and homogenized in 20 mL of ethanol and ultrapure water (80:20, *v/v*) at 4 °C for 15 min using an IKA T-18 Basic Ultra-Turrax homogenizer (IKAWerke GmbH & Co., Staufen, Germany). The homogenate was then centrifuged at 3000× *g* for 10 min at 4 °C, and the supernatant was removed from the residue. The residue was homogenized, and the supernatant removed, for a total of three extractions. The supernatants were then combined and filtered through Whatman No. 1 filter paper.

### 2.4. Total Phenolic Content

The total phenolic (TP) content of leaf extracts was determined using the Folin–Ciocalteu micro method [30]. Three Folin's reactions were made for each olive leaf sample in 1 mL microcentrifuge tubes. The reaction mixture contained 40 μL of extract, 3160 μL of ultrapure water, 200 μL of the Folin–Ciocalteu reagent and 600 μL of 20% sodium carbonate solution. After 30 min of incubation at 40 °C, absorbance was measured at $\lambda$ = 765 nm (UV-1700 Shimadzu, Japan). The TP content is expressed as gallic acid equivalent per dry weight (mg GAE/gdw) for olive leaves. Three independent experiments were performed.

### 2.5. Total Flavonoid Content

Total flavonoid (TF) content was measured using the colorimetric method with aluminum chloride [31,32]. Absorbance was measured at 510 nm and the results were expressed as catechin equivalent per dry weight (mg CE/gdw). Three flavonoid reactions were made for each olive leaf sample.

### 2.6. Determination of Leaf Ionome and Soil Parameters

The soil texture is sandy-clay according to international standards, slightly calcareous, moderately rich in organic matter, phosphorus and potassium, and with a usable water reserve of 1.7 mm.cm$^{-1}$. After excising the petioles, the whole leaves and soil were crushed to a fine powder by a plastic mortar and pestle and sampled at 5 and 10 mg of dry weight. Samples were digested for 1 h at 100 °C in 200 μL of mineral-free concentrated nitric acid. After dilution with ultrapure, mineral-free water and centrifugation at 13,000× *g* to remove any remaining particulates, samples were analyzed by ICP-OES as described by Cobine et al. [21], with simultaneous measurement of Ca, Fe, Mg, Na, Mn, Na, P, S, and Zn. As controls, blanks of nitric acid were digested in parallel. Mineral concentrations were determined by comparing emission intensities to a standard curve created from certified mineral standards (SPEX CertiPrep). Three independent experiments were performed.

### 2.7. Statistical Analyses

Ionome data (individual minerals) were analyzed separately with a one-way analysis of variance (ANOVA) followed by the post-hoc Student–Newman–Keuls test. Principal component analysis was carried out using a correlation matrix. A scatter plot was created according to PC1 and PC2 using SPSS v20 software. SPSS statistical package software (SPSS for Windows, Version 20, SPSS Inc., Chicago, IL, USA) was used for the statistical analysis of data.

## 3. Results

### 3.1. Polymerase Chain Reaction

*X. fastidiosa* PCR detection confirmed that all plant samples analyzed were not infected. No amplified DNA was obtained from any of the tested samples using PCR, confirming the absence of the bacterium in these samples.

### 3.2. Determination of Leaf Ionome

The total concentrations of mineral elements of the leaves sampled are shown in Table 1. The elemental composition was compared with reference values of nutrient content in olive leaves [33]. The concentration of Mg, Mn, Na, and Zn was within these reference ranges. However, Fe and P were considered low or close to the minimal range. Comparing leaf

ionomes, statistical analyses showed that 'Leccino', 'Arbozana', 'Arbiquina', 'Menara' and 'Haouzia' had higher levels of Mn, Cu, Zn, and P. The same varieties also showed lower levels of Ca, Na. Concerning the four remaining varieties, 'Picholine Marocaine', 'Picholine de Languedoc', 'Ogliarola', and 'Cellina di Nardo', data indicated lower levels of Mn, Zn, Cu. These varieties showed higher levels of Ca and Na (Table 1). Nutrient concentrations are expressed in g·kg$^{-1}$, except for Mn, Na, and Zn that are expressed in mg·kg$^{-1}$. Ca concentration was higher than the reference (1–14 mg·kg$^{-1}$).

**Table 1.** Elemental analysis of olive leaves expressed in weight per dry weight. The mean of three replicates was represented.

| | Ca | Mg | P | Cu | Mn | Na | Zn | Fe |
|---|---|---|---|---|---|---|---|---|
| Reference * | mg/kg or g/kg | 10–14 | 1–1.6 | 1–1.3 | - | 20–36 | <200 | 4–9 | 90–124 |
| Arbiquina | NA | 13.20 | 2.04 | 0.82 | 17.35 | 31.18 | 33.30 | 7.80 | 67.05 |
| Arbozana | NA | 12.52 | 2.09 | 0.93 | 18.50 | 35.38 | 28.60 | 10.23 | 67.60 |
| Menara | NA | 13.35 | 2.01 | 0.72 | 17.03 | 31.18 | 35.75 | 7.86 | 69.46 |
| Haouzia | NA | 14.32 | 1.83 | 0.70 | 17.02 | 29.46 | 35.16 | 7.66 | 70.32 |
| Picholine Marocaine | NA | 19.50 | 0.98 | 0.49 | 10.40 | 23.40 | 41.35 | 5.13 | 80.36 |
| Picholine Languedoc | NA | 20.50 | 0.85 | 0.33 | 9.97 | 21.33 | 41.61 | 4.57 | 80.87 |
| Frantoio | NA | 14.45 | 1.75 | 0.68 | 15.25 | 28.82 | 35.05 | 7.25 | 72.34 |
| Leucocarpa | NA | 14.51 | 1.65 | 0.62 | 15.15 | 28.21 | 34.24 | 7.06 | 72.36 |
| Leccino | NA | 7.02 | 3.62 | 2.51 | 23.82 | 42.63 | 20.21 | 15.61 | 41.3 |
| Meslalla | NA | 15.30 | 1.50 | 0.53 | 14.93 | 26.38 | 40.31 | 7.01 | 73.65 |
| Cellina di Nardò | NA | 27.15 | 0.16 | 0.12 | 7.88 | 18.59 | 48.76 | 4.80 | 93.45 |
| Ogliarola | NA | 27.31 | 0.12 | 0.13 | 7.12 | 17.99 | 49.40 | 4.31 | 95.45 |

Element concentrations are expressed in g/kg, except for Fe, Mn, Na, and Zn that are expressed in mg/kg. * Reference concentrations were obtained from Kailis and Harris [34].

Results of soil analysis showed that the soil where olive trees are grown had a higher content in Cu, Zn, and Mn and a low level of Mg. Ca concentration in 0–35 cm soil depth was lower than the reference (3000 mg·kg$^{-1}$), while Mn concentration was higher than the reference (5–20 g·kg$^{-1}$) (Tables 2 and 3).

**Table 2.** Physical and chemical properties of soil.

| Soil Depth | Sand | Silt | Clay | pH | EC | OM | K2O | P2O5 | CaCo3 |
|---|---|---|---|---|---|---|---|---|---|
| Cm | % | % | % | | mS/cm | % | mg·kg$^{-1}$ | mg·kg$^{-1}$ | % |
| 0–35 | 46.8 ± 0.4 | 10.20 | 43.00 | 6.50 | 0.10 | 2.50 | 458.80 | 73.30 | 2.70 |
| 35–70 | 46.10 | 16.10 | 37.60 | 7.80 | 0.10 | 1.60 | 222.50 | 15.10 | 3.10 |

EC: electrical conductivity, OM: organic matter, K2O: potassium, P2O5: phosphorus, CaCo3: calcium carbonates. Data represent means ± standard deviation.

**Table 3.** Chemical analysis of soils wherein the leaf ionome profiles were evaluated.

| | Mg | Cu | Mn | Na | Zn |
|---|---|---|---|---|---|
| Soil analysis (mg·kg$^{-1}$ Fine Fraction) | 237.6 | 19 | 25.3 | 2109 | 0.8 |

Values represent averages of three replicate samplings. All varieties refer to soil collected from the same area.

### 3.3. Determination of Total Phenolic and Flavonoid Content

The total phenolic content in leaves of the studied varieties is reported in Figure 1. Regarding results, the total phenolic content of all varieties varied considerably since several

varieties showed different statistically significant values (F = 157.69, *p* = 0.02). 'Leccino' presented the statistically significant higher phenolic content (45.8 mg·GAE/g) followed by 'Arbiquina', 'Arbozana', 'Menara', 'Haouzia', 'Frantoio', 'Leucocarpa', 'Meslalla', 'Picholine Marocaine', 'Picholine de Languedoc', 'Cellina di Nardò', and 'Ogliarola' which showed the statistically significant lower phenolic content (8.07 mg·GAE/g). Concerning the total flavonoid content, statistical analyses showed a significant difference regarding the variety's values (F = 136.45, *p* = 0.03). 'Leccino' showed the highest total flavonoid content (24.49 mg·GAE/g) and 'Ogliarola' the lowest total flavonoid content (4.89 mg·GAE/g) (Figure 2).

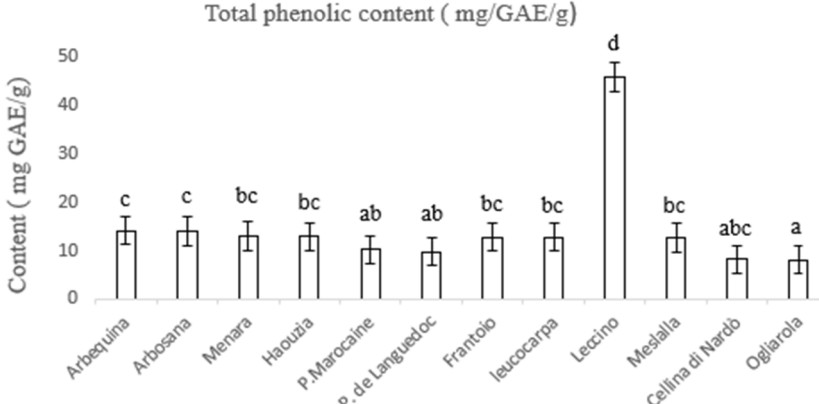

**Figure 1.** Total phenolic content for the studied Olive varieties (mg GAE/g).

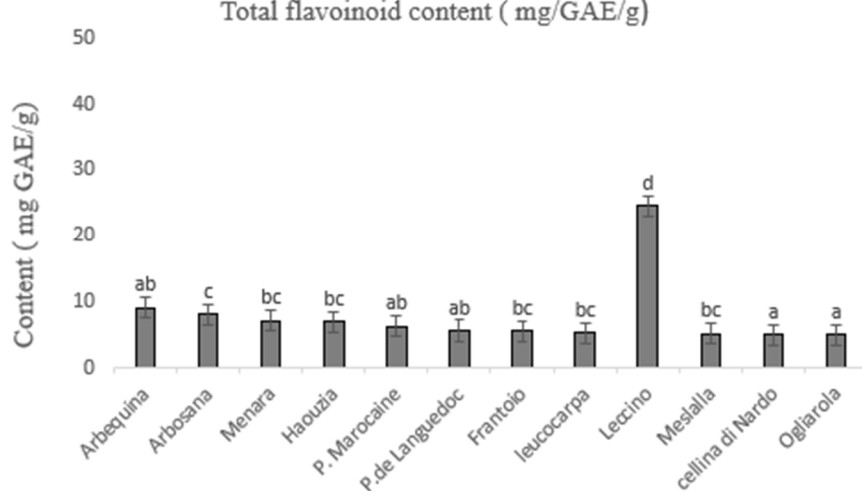

**Figure 2.** Total flavonoid content for the studied Olive varieties (mg GAE/g).

### 3.4. Principal Component Analysis

Principal component analysis (PCA) was used to determine the most significant descriptors in the data set. Only a principal component loading of more than 0.5 was considered as being significant for each factor. Thus, a total variance of 98.47% was explained by only two components. The first component consisted of 10 variables, which represent more than 90% of all total variables, and explained 91.4% of the total variance (Figure 3). The first component accounted for 91.4% of total variance, which is strongly correlated to Ca (r = −0.921), Mn (r = 0.979), Mg (r = 0.987), Na (r = 0.970), Zn (r = 0.974), Cu (r = 0.963), P (r = 0.965), Fe (r = −0.989), TPC (r = 0.939), TFC (0.865). The second function accounted for 7.06% of total inertia.

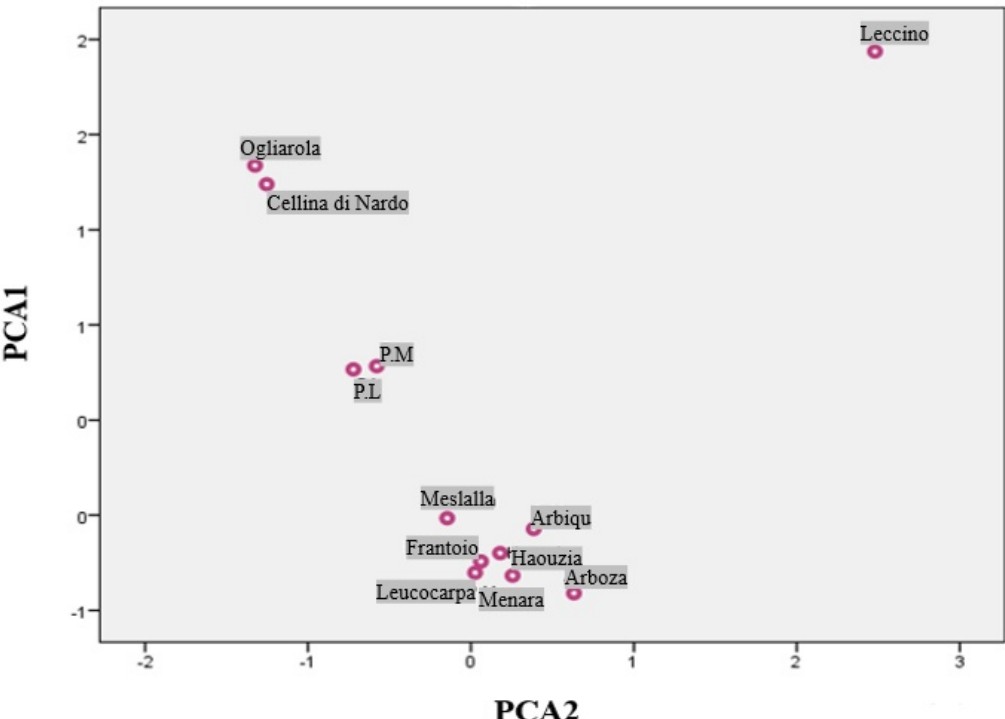

**Figure 3.** Scatter plot for the two principal components (PC1/PC2, 98.47% of total variance) for the studied Olive varieties based on all descriptors.

## 4. Discussion

*Xylella fastidiosa* continues to emerge as a major and devastating bacterial pathogen for innumerable crops, and no cure has been identified so far. Current management strategies are based on the use of cultivars showing resistance to the pathogen in the field such as 'Leccino' [13] and the control of vectors to limit the disease spread. The first requires including tolerant genotypes in breeding programs or replacing susceptible varieties in the current fields [35–37]. Many characteristics make *X. fastidiosa* attractive for studying the effects of nutrients in plant-pathogen interactions. Specifically, since it is xylem-limited, this bacterium is found in the vessels where mineral elements are translocated throughout the plant. Consequently, we spotlight the hypothesis that differences within the ionome of non-infected Moroccan olive trees, the total phenolic and flavonoid content can give an overview of the plasticity and immunity of the Moroccan olive sector to *X. fastidiosa* before the bacterium enters Moroccan groves [36,38].

The outcome of the present investigation shows that 'Leccino' olive variety has a higher Mn, Cu, Zn, total phenolic, and flavonoid content, and lower Ca and Na content which is in line with other previous work [13,14,39]. Manganese, Cu, and Zn are essential micronutrients for plant growth; Mn is involved in the photosynthetic machinery and in the detoxification of reactive oxygen species (ROS) [40], Cu is essential for the formation of chlorophyll [41], and Zn is involved as a cofactor in many enzymes such as alcohol dehydrogenase, carbonic anydrase, and RNA polymerase [42]. It is worth noting that these ions are also strongly involved in the plant defense machinery and in the *X. fastidiosa* virulence. Furthermore, these ions are strongly involved in the plant defense artillery against infections, including *X. fastidiosa* [43]. Specific attention is attributed to Zn ability to reduce the pathogenicity of pathogens [44]. A Zn-finger protein gene, CAZFP1, encodes a zinc-finger transcription factor that builds up in the preliminary phase of the infection of *Xanthomonas campetsris pv. vesicatoria* to pepper fruits [45]. In addition, Zn-fingers binding domains are related to the effector-triggered immune response [46]. High Zn concentrations can preserve plants by direct toxicity and by Zn-triggered organic defenses [43,47]. This confirms the importance of crop nutrient management for a sustainable agriculture [37]. *X. fastidiosa*

biofilm formation is prohibited by Zn and Cu concentrations higher than 0.25 mM, and 200 μM respectively [21], and *in planta*, Zn detoxification is needed to trigger the full virulence of the pathogen [19]. Within this context, previous studies have shown that the supply to the olive canopy of a zinc-copper–citric acid biocomplex, namely Dentamet®, reduces both the field symptoms and *X. fastidiosa* subsp. *pauca* cell densities in the foliage allowing the trees to survive the infection [48]. Recently a high Mn leaf content would appear to match up with a relative level of tolerance in Leccino cultivar to *X. fastidiosa* subsp. *pauca* [13]; the present study would corroborate this feature since both 'Ogliarola salentina' and 'Cellina di Nardò' cultivars are characterized by a lower Mn content than Leccino. The Mn ion is involved in flavonoid and lignin production thereby preserving the cultivar from infection by *X. fastidiosa* subsp. *pauca* [49]. Another important ion, Ca, seems to influence biofilm formation by both extracellular ionic bridging and intracellular stimulation that relies on protein [13]. Ca increases cell attachment probably via type I pili, twitching motility and cell-to-cell attachment responsible for cell aggregation [50]. Consequently, it is a limiting factor in the initial stages of biofilm formation characterized by cell attachment, while it has a less prominent role in late stages of biofilm maturation [15].

Based on those facts, it is logical that olive varieties exhibiting high content of Mn, Cu, and Zn and low content of Ca and Na ('Arbequina', 'Arbosana', 'Menara' and 'Haouzia') would likely be more effective in resisting the development of the infection after the formation of *X. fastidiosa* biofilm. On the other hand, the varieties showing a high content of Ca and Na and reduced content of Mn, Cu, and Zn ('Frantoio', 'leucocap' 'Meslala') would be more adapted to fight the formation of *X. fastidiosa* biofilm in the first place. We also believe that olive varieties with deficiency of the mentioned ions would likely be extremely sensitive and prone to a fast and strong *X. fastidiosa* infection ('Picholine marocaine', and 'Picholine de Languedoc') based on the current data.

Besides ions, phenolic and flavonoid compounds are known for their strong antibacterial effect and their importance in the protection of plants against infections [48,49,51]. The mechanisms of antibacterial action of phenolic compounds are not yet fully deciphered but these compounds are known to involve many sites of action at the cellular level. While phenolic acids have been shown to disrupt membrane integrity, as they cause consequent leakage of essential intracellular constituents [52]. Flavonoids may link to soluble proteins located outside the cells and with bacteria cell walls thus promoting the formation of complexes [53]. Flavonoids also may act through inhibiting both energy metabolism and DNA synthesis thus affecting protein and RNA syntheses. In this study, 'Leccino' had the highest phenolic and flavonoid content, followed by 'Arbequina' and 'Arbozana', and the lowest values were observed in 'Ogliarola' and 'Cellina di Nardò', which is in concordance with a previous study [54]. The high phenolic and flavonoid content may be an indicative parameter on the ability of olive trees to fight *X. fastidiosa* infection based on other studies [25]. Previous studies on grapevine reported an increase of phenolic compounds following *X. fastidiosa* infection, since this plant possesses the ability to change its metabolism towards an excessive formation of phenolic compounds as a defense mechanism against pathogens [25,53,55]. In fact, and following *X. fastidiosa* infection, phenolic compounds (e.g., catechin and digalloylquinic acid), glycosides (e.g., astringin) and flavonoids (e.g., catechins, pyrocyanidins) were found in higher quantities around xylem tissues, where they help the xylem sap to rise and reach different parts of the plant [25]. Thus, those secondary metabolites are determinant in the defense against *X. fastidiosa* infection, and olive varieties with the highest content would, therefore, be more prominent to resist this pathogen.

## 5. Conclusions

The plant pathogen *X. fastidiosa* responsible for the olive quick decline syndrome is considered a quarantine pathogen, and its introduction is highly prohibited in Morocco. This study provided insights on olive varieties that could resist, or are entirely sensitive to *X. fastidiosa* infection, based on the analysis of ions, phenolic and flavonoid content.

Therefore, and as a prevention strategy, we highly recommend increasing the plantation of relatively resistant varieties to *X. fastidiosa* and reducing varieties that could be easily damaged, especially that olive plantation in Morocco contributes to a huge part of its agriculture and economy. However, and since it is only a matter of time until *X. fastidiosa* infects olive trees in Morocco, extensive studies on developing an effective cure are needed. A promising target would be the inhibition of the attachment of *X. fastidiosa* to vegetal cells (e.g., the effect of Ca), inhibiting the first stages of its invasion.

**Author Contributions:** Conceptualization, K.E.H. and E.H.A.; methodology, K.E.H.; software, K.E.H. and M.O.; validation E.H.A. and M.H.; formal analysis, K.E.H.; investigation, E.H.A.; resources, E.H.A.; data curation, K.E.H. and M.S.; writing—original draft preparation, K.E.H.; writing—review and editing, M.E.M.; visualization, K.H. and A.B.; supervision, E.H.A. and M.H.; project administration, Cure-Xf. All authors have read and agreed to the published version of the manuscript.

**Funding:** This research was funded by CURE-XF, an EU-funded project.

**Institutional Review Board Statement:** Not applicable.

**Informed Consent Statement:** Informed consent was obtained from all subjects involved in the study.

**Acknowledgments:** This research was supported by CURE-XF, an EU-funded project, coordinated by CIHEAM Bari (H2020-Marie Sklodowska-Curie Actions-Research and Innovation Staff Exchange. Reference number: 634353). The authors greatly appreciate the English Language editing and review services supplied by Elvira Lapedota, language consultant (https://www.linkedin.com/in/elvira-lapedota-12a53473/) (accessed on 5 May 2021).

**Conflicts of Interest:** The authors declare no conflict of interest.

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
