# Peer review of "Assessment of Ionomic, Phenolic and Flavonoid Compounds for a Sustainable Management of Xylella fastidiosa in Morocco"

_sustainability, doi:10.3390/su13147818_

Round 1
Reviewer 1 Report
Very interesting study, however there are some concerns:
- english language checking from a native speaker is needed.
- -I suggest authors to avoid reporting as keywords words already written in the title.
- scientific name of species and cultivar to be written in italic
- check for measure unit formatting
Author Response
Point 1: English language checking from a native speaker is needed.
Point 2: I suggest authors to avoid reporting as keywords words already written in the title.
Point 3: Scientific name of species and cultivar to be written in italic
Point 4: Check for measure unit formatting
Response 1: The English Language editing and review services was supplied by Elvira Lapedota, language consultant (https://www.linkedin.com/in/elvira-lapedota-12a53473/)
Response 2: Current Keywords: Olive quick decline syndrome; Xylella fastidiosa; Calcium; Manganese; Leccino; Ogliarola salentina; Olive; Moroccan olive varieties; Mediterranean olive varieties.
Response 3: We checked that all Scientific names of species and cultivar are written in italic.
Response 4: mg.kg-1 was replaced by mg.kg-1 and
Reviewer 2 Report
Assessment of Ionomic, Phenolic and Flavonoid Compounds in Xylella fastidiosa Non-Infected Olive leaves in Morocco.
The authors investigate Olive cultivars to determine which are likely to be resistant to the bacteria Xylella fastidiosa and apply a scale to which cultivars are likely resistant to most susceptible.
The manuscript appears to be scientifically sound and the topic is interesting. However, the current state of the manuscript needs some additional information and organization. The English language needs some work and the style is informal in some places.
1) Additional information about the pathogen is needed in the Introduction. The authors state it is a bacteria but other than that there is no background information on the pathogen. What kind of bacteria, what host species does it infect, current geographical range, ect.
For example: Abstract, the authors state " X. fastidiosa leads to devastating diseases and important economic losses."
Diseases implies it causes more than one disease (which is true) but there is no mention of the extent of this in the introduction. It is not off topic and will provide the reader with a greater understanding of the pathogen and support the why the research was completed.
2) Organization: Unless I missed it, Table 3 is discussed before both Tables 1 and 2. If this is the case, Table 3 needs to be Table 1, and Table 1 and Table 2 needs to be adjusted. Some sentences in the results appear to be interpretation and need to be removed. In the results, please just state the results.
3) All Genus species names should be in italics.
Line items:
Abstract:
...along with power tool for the management...
Consider revising: ...along with being a powerful tool for management...
The findings suggest that...
Consider revising to something like: Our findings suggest that 'Arbozana', 'Arbiquina', 'Menara' and 'Haouzia' may tolerate the infection by X. fastidiosa to varying degrees, confirms that 'Leccino' has high resistance to X. fastidiosa, and both 'Ogliarola salentina' and 'Cellina di Nardò' are sensitive to X. fastidiosa infection.
Key Words:
Please remove all key words that are in the title and add different words.
Introduction:
Page 1
1) ... they hand out to the maintenance of the rural populations.
Consider revising as this is awkward as written.
2) It represents the Southwestern extreme...
Consider revising to something like: Moroccan olive plantations represent the Southwestern extreme...
3) Furthermore, over the last years, Moroccan olive area has...
Consider revising to something like: Furthermore, in recent years land use for cultivating olives in Morocco has increased...
4) ...is the utmost predominant variety;
Consider removing utmost as it is redundant with predominant.
Page 2
5) Top of page: Need a period after clonal selection [4].
6)... Morocco is located at just 13 km away from Spain...
Consider revising: ... Morocco is 13 km from Spain...
7) Is there extra spacing before and after mineral nutrients
8) ...inter-connection with the pathogen infection [14].
Consider revising as it is awkward as written.
9) It was very recommended that the host ionome and its variation could be considered...
This is wordy as written, consider revising.
10) ...in vitro growth...
I believe in vitro should be in italics.
Materials and Methods
Page 3
Polymerase chain reaction
11) Thermocycling conditions were like so:...
This is too informal, consider revising: PCR was completed using a XXXX thermocyler set to the following:
12) Electrophoresis is a visualization tool, not an analytical test.
Consider revising to something like: The resulting PCR products were visualized by electrophoresis in 1% Tris-Acetate-EDTA agarose gel stained with ethidium bromide.
13) Need to capitalize the first word of each sentence:
Xie and Bolling [29]. Primary,
This is also awkward as written: Consider revising: First, 1 g aliquots of every powder were...
Also, please describe the process of obtaining this powder. The methods should be stand alone.
For example: All leaves were ground into a powder using xxxx, placed into polypropylene tubes, and homogenized in ....
Please add all the steps for the experiment can be repeated.
14) supernatant removed as above...
Please clarify.
Page 4
Statistical analyses
15) Capitalize: A scatter plot was...
Results
Page 4
Polymerase chain reaction
16) The use of primers....
All of this should go in the methods as it is not a result but a justification for using those primers.
Determination of leaf ionome
17) Comparing ionomes....
All sentences below with numbers seem redundant if this information is in a table. If not, a table would better represent this information.
18) First mention of Table 3.
Table 1 and 2 first mention is on page 5. This means Table 3 is actually Table 1. Tables are numbered in order of mention. This needs adjusted.
Page 5
19) All tables are replicates of 3? If so why are there so few mean without stand deviations?
20) Table 3: The format for g/kg or mg/kg needs adjusted.
Page 6
21) Determination of total phenolic and flavonoid content.
This section needs to be reworded to be more concise.
22) Figure 1 and Figure 2. Should Total be capitalized in both?
Page 7
Principal Component Analysis
23) Reason why these attributes had...
24) Taking into account...
Both sentence are confusing as written, please reword.
Page 8
25) Consequently, we spotlight....
Please reword the entire sentence as it is awkward.
26) Zinc is deeply involved in the mechanisms of plant defense in relation to [41].
Please complete in ration to.......and add [41] at the end. There is no context to the sentence as written.
27) Is there an extra space after [42].
28) in planta...
Should this be in italics?
29) Recently a high Mn leaf content would...
Please reword as it is awkward as written.
30) This ion is involved...
Please just state the ion... The ion Mn is involved...
31) ....occupy second place..
32) In third place...
33) In fourth place....
This is informal and should be reworded.
34) Our data were similar to those.....
Please reword the entire sentence and the next sentence: awkward as written.
Conclusions
Page 9
35) is prohibited.
Please reword as the bacteria is not prohibited to enter.
36) farmers to face this alien bacterium entered Moroccan groves.
Please reword, awkward as written.
37) ...and skip the danger that may face...
Please reword, awkward as written.
Please note that there are additional location within the manuscript that need to be reword so please review the entire manuscript.
Author Response
1) Additional information about the pathogen is needed in the Introduction. The authors state it is a bacteria but other than that there is no background information on the pathogen. What kind of bacteria, what host species does it infect, current geographical range, ect.
For example: Abstract, the authors state " X. fastidiosa leads to devastating diseases and important economic losses."
Diseases implies it causes more than one disease (which is true) but there is no mention of the extent of this in the introduction. It is not off topic and will provide the reader with a greater understanding of the pathogen and support the why the research was completed.
Response: we added a paragraph where we mention additional backgroud information about the pathogen
3) All Genus species names should be in italics.
Response: All Genus species are now in italics
Line items:
Abstract:
...along with power tool for the management...
Consider revising: ...along with being a powerful tool for management...
Response: The assessment of the macro and micro-elements content (ionome) in leaves can give basic and useful information along with being a powerful tool for management of diseases caused by this devastating pathogen.
The findings suggest that...
Consider revising to something like: Our findings suggest that 'Arbozana', 'Arbiquina', 'Menara' and 'Haouzia' may tolerate the infection by X. fastidiosa to varying degrees, confirms that 'Leccino' has high resistance to X. fastidiosa, and both 'Ogliarola salentina' and 'Cellina di Nardò' are sensitive to X. fastidiosa infection.
Response: Our findings suggest that 'Arbozana', 'Arbiquina', 'Menara' and 'Haouzia' may tolerate the in-fection by X. fastidiosa to varying degrees, confirm that 'Leccino' has high resistance to X. fastidiosa, and both 'Ogliarola salentina' and 'Cellina di Nardò' are sensitive to X. fastidiosa infection.
Key Words:
Please remove all key words that are in the title and add different words.
Response: the new keywords are: Olive quick decline syndrome; Xylella fastidiosa; Calcium; Manganese; Leccino; Ogliarola salentina; Olive; Moroccan olive varieties; Mediterranean olive varieties
Introduction:
Page 1
1) ... they hand out to the maintenance of the rural populations.
Response 1: In Morocco, olive (Olea europaea subsp. europaea L.) groves have a crucial socio-economic role, representing the main source of livelihood for many local farmers
2) It represents the Southwestern extreme...
Response 2: Moroccan olive groves represent the South-westernmost part of the Mediterranean olive-growing landscape. In this country, olive cultivation and oil production are a deep-rooted tradition, both as an income for more than 450 000 farmers and a high environmental value crop, due to its role in soil protection, particularly, on mountain farms.
3) Furthermore, over the last years, Moroccan olive area has...
Consider revising to something like: Furthermore, in recent years land use for cultivating olives in Morocco has increased...
Response 3: Furthermore, over the last few years, land use for olive cultivation in Morocco has increased from 946 818 ha in 2014 to 1 073 493 ha in the 2019 growing season [2] making this crop one of the most profitable and strategic horticultural crops in the country
4) ...is the utmost predominant variety;
Consider removing utmost as it is redundant with predominant.
Response 4: The ‘Picholine Marocaine’ is the predominant variety; more than 96% of olive groves are planted with this variety
Page 2
5) Top of page: Need a period after clonal selection [4].
Response 5: Done
6)... Morocco is located at just 13 km away from Spain...
Consider revising: ... Morocco is 13 km from Spain...
Response 6: Indeed, Morocco is 13 km from Spain and is the gateway to all African countries. Also, Morocco has significant trade with several European countries, notably France, Spain, and Italy, which may contribute to the introduction of X. fastidiosa indirectly via infected plant material.
7) Is there extra spacing before and after mineral nutrients
Response 7: The extra spacing was removed
8) ...inter-connection with the pathogen infection [14].
Consider revising as it is awkward as written.
Response 8: The analysis of the whole profile of trace elements and mineral nutrients can contribute to evaluate the physiological status of the plant in inter-connection with the pathogen infection.
9) It was very recommended that the host ionome and its variation could be considered...
This is wordy as written, consider revising.
Response 9: It was highly recommended that the host ionome and its variation could be considered as a potential tool for the control of diseases caused by this xylem-limited phytopathogenic bacterium [13]. Mineral transport and balance are crucial for the growth and development of plants and also microorganisms and can be a major factor in disease control and progression.
10) ...in vitro growth...
I believe in vitro should be in italics.
Response10: Done
Materials and Methods
Page 3
Polymerase chain reaction
11) Thermocycling conditions were like so:...
This is too informal, consider revising: PCR was completed using a XXXX thermocyler set to the following:
Response 11: PCR was completed using a (5 PRIMEG/C Serial No 51147-2) thermocyler set to the following: 94 °C for 3 min, 35 cycles of 94°C for 30 sec, 50-55 °C for 30-45 sec and 72 °C for 30 sec, and a final extension of 5 min at 72 °C [26].
12) Electrophoresis is a visualization tool, not an analytical test.
Consider revising to something like: The resulting PCR products were visualized by electrophoresis in 1% Tris-Acetate-EDTA agarose gel stained with ethidium bromide.
Response 12: The resulting PCR products were visualized by electrophoresis in 1% Tris-Acetate-EDTA agarose gel stained with ethidium bromide.
13) Need to capitalize the first word of each sentence:
Xie and Bolling [29]. Primary,
Response 13: by Xie and Bolling [29]. First, 1 g aliquots of powder were transferred into polypropylene tubes and homogenized in 20 ml of ethanol and ultrapure water (80:20, v/v) at 4 °C for 15 min using an IKA T-18 Basic Ultra-Turrax homogenizer (IKAWerke GmbH & Co., Staufen, Germany) This is also awkward as written: Consider revising: First, 1 g aliquots of every powder were...
Also, please describe the process of obtaining this powder. The methods should be stand alone.
For example: All leaves were ground into a powder using xxxx, placed into polypropylene tubes, and homogenized in ....
Response: Then, they were ground into powder at room temperature using an IKA A11 Basic Grinder (St. Louis, MO, USA). Extraction was based on the method previously described by Sanders et al.
(Description of the method is in the reference)
14) supernatant removed as above...
Please clarify.
Response14: The residue was homogenized, and the supernatant removed, for a total of three extractions. The supernatants were then combined and filtered through Whatman No. 1 filter paper.
Page 4
Statistical analyses
15) Capitalize: A scatter plot was...
Response 15: Done
Results
Page 4
Polymerase chain reaction
16) The use of primers....
All of this should go in the methods as it is not a result but a justification for using those primers.
Response 16: Xylella fastidiosa PCR detection confirmed that all plant samples analyzed were not infected. No amplified DNA was obtained from any of the tested samples using PCR, confirming the absence of the bacterium in these samples.
Determination of leaf ionome
17) Comparing ionomes....
All sentences below with numbers seem redundant if this information is in a table. If not, a table would better represent this information.
Response 17: Sentences were removed
18) First mention of Table 3.
Table 1 and 2 first mention is on page 5. This means Table 3 is actually Table 1. Tables are numbered in order of mention. These needs adjusted.
Response 18: that’s true, now tables are numbered in order of mention
Page 5
19) All tables are replicates of 3? If so, why are there so few means without stand deviations?
Response 5: Elemental analysis of olive leaves expressed in weight per dry weight. The mean of three replicates was represented.
20) Table 3: The format for g/kg or mg/kg needs adjusted.
Response 20: mg.kg-1 is replaced by mg.kg-1
Page 6
21) Determination of total phenolic and flavonoid content.
This section needs to be reworded to be more concise.
Response 21: The total phenolic content in leaves of the studied varieties is reported in Figure 1. Regarding results, the total phenolic content of all varieties varied considerably since several varieties showed different statistically significant values (F=157.69, p=0.02). ‘Leccino’ presented the statis-tically significant higher phenolic content (45.8 mg GAE/g) followed by ‘Arbiquina’, ‘Arbozana’, ‘Menara’, ‘Haouzia’, ‘Frantoio’, ‘Leucocarpa’, ‘Meslalla’, ‘Picholine Marocaine’, ‘Picholine de Languedoc’, ‘Cellina di Nardò’, and ‘Ogliarola’ which showed the statistically significant lower phenolic content (8.07 mg GAE/g). Concerning the total flavonoid content, statistical analyses showed a significant difference regarding the variety’s values (F= 136.45, p=0.03). ‘Leccino’ showed the highest total flavonoid content (24.49 mg GAE/g) and ‘Ogliarola’ the lowest total flavonoid content (4.89 mg GAE/g).
22) Figure 1 and Figure 2. Should Total be capitalized in both?
Response 22: True, now both are capitalized
Page 7
Principal Component Analysis
23) Reason why these attributes had...
24) Taking into account...
Both sentences are confusing as written, please reword.
Response 23 and 24: This is the reason why these attributes had the highest variation between varieties and a great impact on their discrimination. The first component accounted for 91.4% of total variance, which is strongly correlated to Ca (r= -0,921), Mn (r=0,979), Mg (r=0,987), Na (r=0-,970), Zn (r=0,974), Cu (r=0,963), P (r=0,965), Fe (r=-0,989), TPC (r=0,939), TFC (0,865) Taking into account the number of the evaluated accessions, it is estimated that these correlations are highly strong and significant. The second function accounted for 7.06% of total inertia.
Page 8
25) Consequently, we spotlight....
Please reword the entire sentence as it is awkward.
Response 25: Consequently, we spotlight the hypothesis that differences within the ionome of non-infected Moroccan olive trees, the total phenolic and flavonoid content can give an overview of the plasticity and immunity of the Moroccan olive sector to X. fastidiosa before the bacterium enters Moroccan groves.
26) Zinc is deeply involved in the mechanisms of plant defense in relation to [41].
Please complete in ration to.......and add [41] at the end. There is no context to the sentence as written.
Response 26: Zn is involved as a cofactor in many enzymes such as alcohol dehydrogenase, carbonic anydrase, and RNA polymerase [40]. It is worth noting that these ions are also strongly involved in the plant defense machinery and in the X. fastidiosa virulence. Furthermore, these ions are strongly involved in the plant defense artillery against infections, including X. fastidiosa [39]. Specific at-tention is attributed to Zn ability to reduce the pathogenicity of pathogens
27) Is there an extra space after [42].
Response 27: Extra space is removed
28) in planta...
Should this be in italics?
Response28: Yes, that’s True
29) Recently a high Mn leaf content would...
Please reword as it is awkward as written.
Response 29: Recently a high Mn leaf content would appear to match up with a relative level of tolerance in Leccino cultivar to X. fastidiosa subsp. pauca [13]; the present study would corroborate this feature since both ‘Ogliarola salentina’ and ‘Cellina di Nardò’ cultivars are characterized by a lower Mn content than Leccino.
30) This ion is involved...
Please just state the ion... The ion Mn is involved...
Response 30: The Mn ion is involved in flavonoid and lignin production thereby pre-serving the cultivar from infection by X. fastidiosa subsp. pauca
31) ....occupy second place..
32) In third place...
33) In fourth place....
This is informal and should be reworded.
Response 31,32 and 33: Based on those facts, it is logical that olive varieties exhibiting high content of Mn, Cu, and Zn and a low content of Ca and Na (‘Arbequina’, ‘Arbosana’, ‘Menara’ and ‘Haouzia’) would be more effective in resisting the development of the infection after the formation of X. fastidiosa biofilm. On the other hand, the varieties showing a high content of Ca and Na and reduced content of Mn, Cu, and Zn (‘Frantoio’, ‘leucocap’ ‘Meslala’) would be more adapted to fight the formation of X.fastidiosa biofilm in the first place
34) Our data were similar to those.....
Please reword the entire sentence and the next sentence: awkward as written.
Response 34: Previous studies on grapevine reported an increase of phenolic compounds following X.fastidiosa infection, since this plant possesses the ability to change its metabolism towards an excessive formation of phenolic compounds as a defense mechanism against pathogens
Conclusions
Page 9
35) is prohibited.
Please reword as the bacteria is not prohibited to enter.
36) farmers to face this alien bacterium entered Moroccan groves.
Please reword, awkward as written.
37) ...and skip the danger that may face...
Please reword, awkward as written.
Response 35/36/37: The plant pathogen X.fastidiosa responsible for the olive quick decline syndrome is considered as a quarantine pathogen, and its introduction is highly prohibited in Morocco. This study pro-vided insights on olive varieties that could resist, or are entirely sensitive to X.fastidiosa infection, based on the analysis of ions, phenolic and flavonoid content. Therefore and as a prevention strategy, we highly recommend to increase the plantation of relatively resistant varieties to X.fastidiosa and to reduce varieties that could be easily damaged, especially that olive plantation in Morocco contributes to a huge part of its agriculture and economy. However and since it is only a matter of time until X.fastidiosa infects olive trees in Morocco, extensive studies on developing an effective cure are needed. A promising target would be the inhibition of the attachment of X.fastidiosa to vegetal cells (e.g. effect of Ca), inhibiting the first stages of its invasion.
Please note that there is additional location within the manuscript that need to be reword so please review the entire manuscript.
Response: The English Language editing and review services was supplied by Elvira Lapedota, language consultant (https://www.linkedin.com/in/elvira-lapedota-12a53473/)
Round 2
Reviewer 2 Report
The manuscript is greatly improved and the organization is great. The main concern at this point is the overall tone of the manuscript which needs to be adjusted in a few places. This is because the authors did not directly test for resistance or sensitivity. I agree that it is likely the ionome is an indication, but there are other factors that may equate to resistance vs sensitive. Without experimentally infecting the olives to determine which varieties are resistance vs sensitive, I believe less definitive statements are warranted. Most of my line items below relate to this point.
1) In the Discussion (page 8) the authors state: Current management strategies are based on the use of cultivars showing resistance to the pathogen in the field such as ' Leccino' and the control of vectors to limit the disease spread. If there is published data that ' Leccino' is resistance, then please add the reference to this statement. If not, and the addition of such as ' Leccino' is based on this study, this should be removed.
Example 1: Current management strategies are based on the use of cultivars showing resistance to the pathogen in the field such as ' Leccino' [citation] and the control of vectors to limit the disease spread.
or Example 2: Current management strategies are based on the use of cultivars showing resistance to the pathogen in the field and the control of vectors to limit the disease spread.
Abstract: at the end of the abstract:
2) Our findings suggest that 'Arbozana', 'Arbiquina', 'Menara', and 'Haouzia' may tolerate the infection by X. fastidiosa to varying degrees, confirm that 'Leccino' has high resistance to X. fastidiosa, and both ' Ogliarola salentina' and 'Cellina di Nardö' are sensitive to X. fastidiosa infection.
I think this study adds support for the high resistance to X. fastidiosa and support that the last to cultivars are likely sensitive, but without experimentally testing that with the pathogen this is over-reach.
Suggest changing to something like:
Our findings suggest that 'Arbozana', 'Arbiquina', 'Menara', and 'Haouzia' may tolerate the infection by X. fastidiosa to varying degrees, provides addtional support for 'Leccino' having resistance to X. fastidiosa, and suggests that both ' Ogliarola salentina' and 'Cellina di Nardö' are likely sensitive to X. fastidiosa infection.
Results
Principal Component Analysis
3) Two sentences do not appear to be results and should be removed or rewritten. These are:
1) This is the reason why these attributes had....
2) Taking into account the number of evaluated accessions....
Discussion
1) See first statement above related to the discussion.
4) Based on those facts, it is logical that olive varieties exhibiting high content of Mn, Cu, and Zn and a low content of Ca and Na (‘Arbequina’, ‘Arbosana’, ‘Menara’ and ‘Haouzia’) would be more effective in resisting the development of the infection after the formation of X. fastidiosa biofilm.
This is too definitive based on what I stated above.
Consider changing to something like
Based on those facts, it is logical that olive varieties exhibiting high content of Mn, Cu, and Zn and a low content of Ca and Na (‘Arbequina’, ‘Arbosana’, ‘Menara’ and ‘Haouzia’) would likely be more effective in resisting the development of the infection after the formation of X. fastidiosa biofilm.
5) One doubtless fact is that olive varieties with deficiency of the mentioned ions would be extremely sensitive and prone to a fast and strong X. fastidiosa infection (‘Picholine marocaine’, and ‘Picholine de Languedoc’).
This is not doubtless without direct pathogen testing. Please consider softening the sentence to something like:
We also believe that olive varieties with deficiency of the mentioned ions would likely be extremely sensitive and prone to a fast and strong X. fastidiosa infection (‘Picholine marocaine’, and ‘Picholine de Languedoc’) based on the current data.
6) The high phenolic and flavonoid content is an indicative parameter on the ability of olive trees to fight X. fastidiosa infection.
This sentence needs a citation or softened to something like (also not sure if there is a space between X. and fastidiosa):
The high phenolic and flavonoid content may be an indicative parameter on the ability of olive trees to fight X. fastidiosa infection based on other studies [citaion].
Author Response
1) In the Discussion (page 8) the authors state: Current management strategies are based on the use of cultivars showing resistance to the pathogen in the field such as ' Leccino' and the control of vectors to limit the disease spread. If there is published data that ' Leccino' is resistance, then please add the reference to this statement. If not, and the addition of such as ' Leccino' is based on this study, this should be removed.
Example 1: Current management strategies are based on the use of cultivars showing resistance to the pathogen in the field such as ' Leccino' [citation] and the control of vectors to limit the disease spread.
or Example 2: Current management strategies are based on the use of cultivars showing resistance to the pathogen in the field and the control of vectors to limit the disease spread.
Response: Yes, there is published data that ' Leccino' is resistance. So, we choose the first suggestion and we added the reference: Current management strategies are based on the use of cultivars showing resistance to the pathogen in the field such as ' Leccino' [13] and the control of vectors to limit the disease spread.
Abstract: at the end of the abstract:
2) Our findings suggest that 'Arbozana', 'Arbiquina', 'Menara', and 'Haouzia' may tolerate the infection by X. fastidiosa to varying degrees, confirm that 'Leccino' has high resistance to X. fastidiosa, and both ' Ogliarola salentina' and 'Cellina di Nardö' are sensitive to X. fastidiosa infection.
I think this study adds support for the high resistance to X. fastidiosa and support that the last to cultivars are likely sensitive, but without experimentally testing that with the pathogen this is over-reach.
Suggest changing to something like:
Our findings suggest that 'Arbozana', 'Arbiquina', 'Menara', and 'Haouzia' may tolerate the infection by X. fastidiosa to varying degrees, provides addtional support for 'Leccino' having resistance to X. fastidiosa, and suggests that both ' Ogliarola salentina' and 'Cellina di Nardö' are likely sensitive to X. fastidiosa infection.
Response: A good suggests! we changed it to: Our findings suggest that 'Arbozana', 'Arbiquina', 'Menara', and 'Haouzia' may tolerate the infection by X. fastidiosa to varying degrees, provides addtional support for 'Leccino' having resistance to X. fastidiosa, and suggests that both ' Ogliarola salentina' and 'Cellina di Nardö' are likely sensitive to X. fastidiosa infection.
Results
Principal Component Analysis
3) Two sentences do not appear to be results and should be removed or rewritten. These are:
1) This is the reason why these attributes had....
2) Taking into account the number of evaluated accessions....
Response: The two sentences are removed.
Discussion
1) See first statement above related to the discussion.
Response: Done
4) Based on those facts, it is logical that olive varieties exhibiting high content of Mn, Cu, and Zn and a low content of Ca and Na (‘Arbequina’, ‘Arbosana’, ‘Menara’ and ‘Haouzia’) would be more effective in resisting the development of the infection after the formation of X. fastidiosa biofilm.
This is too definitive based on what I stated above.
Consider changing to something like
Based on those facts, it is logical that olive varieties exhibiting high content of Mn, Cu, and Zn and a low content of Ca and Na (‘Arbequina’, ‘Arbosana’, ‘Menara’ and ‘Haouzia’) would likely be more effective in resisting the development of the infection after the formation of X. fastidiosa biofilm.
Response: we changed it according to your suggestion
5) One doubtless fact is that olive varieties with deficiency of the mentioned ions would be extremely sensitive and prone to a fast and strong X. fastidiosa infection (‘Picholine marocaine’, and ‘Picholine de Languedoc’).
This is not doubtless without direct pathogen testing. Please consider softening the sentence to something like:
We also believe that olive varieties with deficiency of the mentioned ions would likely be extremely sensitive and prone to a fast and strong X. fastidiosa infection (‘Picholine marocaine’, and ‘Picholine de Languedoc’) based on the current data.
Response: True, this is not doubtless without direct pathogen testing, we changed it according to your suggestion
6) The high phenolic and flavonoid content is an indicative parameter on the ability of olive trees to fight X. fastidiosa infection.
This sentence needs a citation or softened to something like (also not sure if there is a space between X. and fastidiosa):
The high phenolic and flavonoid content may be an indicative parameter on the ability of olive trees to fight X. fastidiosa infection based on other studies [citaion].
Response: we added the reference: The high phenolic and flavonoid content may be an indicative parameter on the ability of olive trees to fight X. fastidiosa infection based on other studies [25].